# A Systematic Review of Fumagillin Field Trials for the Treatment of Nosema Disease in Honeybee Colonies

**DOI:** 10.3390/insects15010029

**Published:** 2024-01-02

**Authors:** Michael Peirson, Stephen F. Pernal

**Affiliations:** Agriculture and Agri-Food Canada, Beaverlodge Research Farm, P.O. Box 29, Beaverlodge, AB T0H 0C0, Canada

**Keywords:** fumagillin, dicyclohexylamine, *Nosema apis*, *Nosema ceranae*, antibiotic resistance, *Vairimorpha*

## Abstract

**Simple Summary:**

Fumagillin is an antibiotic used to control nosema disease in honeybees, but some authors have suggested that it is no longer effective. This review summarizes the findings of 50 controlled field trials to determine whether fumagillin inhibits both common species of *Nosema*, whether the effectiveness of fumagillin has changed over time, and whether the use of fumagillin brings benefit or harm to the honeybee colony. Fumagillin inhibits both *Nosema* species, and the results of field trials provide no evidence that resistance has developed. Fumagillin-treated colonies had fewer infected bees and lower nosema spore counts than untreated colonies. Furthermore, we found no reports that fumagillin caused significant negative effects on honeybee colonies, among the field trials meeting our criteria for review. There were also many reports of improvements in colony survival, size, and productivity with treatment. However, there was inadequate evidence to determine the best conditions, timing, dose, and method of application. This report demonstrates that fumagillin remains a valuable tool to promote honeybee colony health.

**Abstract:**

This article systematically reviews controlled field trials of fumagillin dicyclohexylamine in honeybee colonies to determine whether fumagillin effectively controls nosema and whether it is beneficial to colonies. Fifty publications were found that described controlled field trials of fumagillin in honeybee colonies between 1952 and 2023. Fumagillin consistently reduced the prevalence and severity of nosema infections. Doses applied in recent studies were similar to or below those recommended historically. Furthermore, our study showed no negative effects on colony health. Improvements in colony survival, size, and honey production have been demonstrated frequently, though not consistently, in both historic and recent studies. Nevertheless, some practices are not optimal. Treatment decision thresholds based on the number of spores per bee are not well supported by evidence and may be no better than calendar-based prophylactic treatments. In addition, reasonable recommendations to employ quarantine and disinfection procedures together with fumagillin treatment do not appear to have been widely adopted. When used as stand-alone treatments, both the fall- and spring-label doses provide benefits but may be too low and short-term to ensure full control of the disease.

## 1. Introduction

*Vairimorpha apis* and *Vairimorpha ceranae* (herein referred to by their former names *Nosema apis* Zander and *Nosema ceranae* Fries) [1] are well-known pathogens of honeybee (*Apis mellifera* L.) colonies. Since the 1950s, the antibiotic fumagillin, usually formulated as fumagillin dicyclohexylamine [2,3,4], has been available to treat nosema disease, although the routine use of this antibiotic is not permitted in all countries [5,6]. Fumagillin is produced by the fungus *Aspergillus fumigatus* and targets the methionine aminopeptidase type 2 (MetAP-2) protein. Fumagillin acts against microsporidian and mammalian MetAP-2 enzymes, and this is the cause of its potential toxicity to humans [3]. Fumagillin treatment stops active nosema infections but does not prevent reinfection by spores remaining in the colony, and consequently, treatments are typically applied once or twice per year, outside of the honey production season [3].

Following the discovery that *N. ceranae* can infect *A. mellifera* [7], questions have been raised about the continued value of fumagillin in honeybee colonies. These questions have become more acute by increasingly widespread restrictions on the use of antibiotics in food production [8]. Many recent reviews [5,7,9,10,11,12,13,14,15,16,17] have discussed fumagillin, but there still appears to be doubt about whether fumagillin effectively controls nosema infections, whether it provides a net benefit to honeybee colonies, and whether the effectiveness of this antibiotic has changed over time [10,12].

A systematic review is “a review that uses explicit, systematic methods to collate and synthesize findings that address a clearly formulated question” [18]. Systematic reviews may or may not include formal statistical procedures such as the meta-analysis of effect estimates. While conducting a large field trial involving fumagillin treatment [19], we attempted to identify all previous field studies describing the use of fumagillin in honeybee colonies. This review summarizes observations from controlled field trials conducted between fumagillin’s introduction in 1952 and January 2023. Our objectives were to answer the following questions:Does fumagillin, as applied in honeybee colonies, reduce the prevalence and intensity of both species of *Nosema* commonly found in honeybee colonies?Given that fumagillin has been widely used in honeybee colonies for seventy years, is there evidence that either species of *Nosema* has become less sensitive to fumagillin?Is the use of fumagillin in honeybee colonies, on balance, associated with benefits to the colonies (increased survival, larger colony size, and greater honey production) or with harm (reduced survival, smaller colony size, and reduced honey production)?

## 2. Materials and Methods

This study conforms to PRISMA (Preferred Reporting Items for Systematic Review and Meta-Analysis Protocols) guidelines [18].

### 2.1. Literature Search

Field trials of fumagillin in honeybee colonies were identified using a citation-based search strategy as follows:(1)An initial group of recent articles related to fumagillin and nosema was identified by manually inspecting the titles and abstracts of all articles in the *Journal of Apicultural Research* and the *Journal of Economic Entomology* between 2010 and June 2021. Additional articles were identified from the journal *PLoS One* using keyword searches for *Nosema ceranae*, *Nosema apis*, and fumagillin.(2)Each of the identified articles was examined to determine if it was an original research article describing a field trial with fumagillin treatment. All citations of previous fumagillin field trials or of reviews that might include citations of previous field trials were noted, and these works were added to the list of articles to be reviewed.(3)The citation of each article that was a fumagillin field trial or a review of fumagillin field trials was entered into Google Scholar. Titles and abstracts of subsequent articles that cited the trial or review were manually inspected. If the subsequent article appeared to be a trial involving fumagillin or a review of such trials, the full text of the article was obtained and reviewed in depth.(4)Steps (2) and (3) were repeated iteratively on each identified field trial or related work until no additional field trials were identified. The search process was concluded in January 2023.

### 2.2. Criteria for Inclusion

An experiment was considered a field trial of fumagillin if it (a) was conducted in honeybee colonies, nucleus colonies, or other conditions typical of natural or commercial bee management; (b) included at least one group that was treated with a known quantity of fumagillin; and (c) included at least one group that could reasonably be considered to be an untreated control for the fumagillin treatment. Experiments conducted in laboratory conditions with caged bees and experiments conducted in colonies but without fumagillin treatments at the colony level were excluded. Experiments in which a before-treatment measurement served as the control for the after-treatment measurement of the same colonies were not considered controlled experiments because the results were confounded by time. Experiments that compared different beekeeping operations, some of which applied fumagillin and some of which did not, were excluded because the effects of treatment could not be distinguished from other aspects of management and colony history.

### 2.3. Summaries of Effects

Outcomes of interest were (a) changes in the number of infected bees or colonies (prevalence); (b) changes in the number of spores per bee (intensity); (c) colony survival; (d) adult bee population; (e) brood population; (f) queen survival; (g) honey production; and (h) other hive products. The effect of fumagillin was summarized by placing each outcome of each trial into one of five categories as follows: (1) beneficial and statistically significant (compared with controls); (2) beneficial but not statistically significant (or no statistical test reported); (3) neutral; (4) harmful but not statistically significant (or no statistical test reported); and (5) harmful and statistically significant. For nosema measurements, “beneficial” meant that the nosema prevalence or intensity was lower in the fumagillin-treated group. For colony characteristics, “beneficial” meant greater colony populations, more honey and hive product production, and/or increased survival. Significance tests were those reported by the original authors. Nonsignificant results were considered neutral if the reported value for the fumagillin-treated group was within 10% of the value of the control group. For survival measurements, nonsignificant results were considered neutral unless the number of deaths in the fumagillin-treated group differed by more than 10% and by at least five colonies, compared with the expected number based on deaths in the control group.

### 2.4. Differences among Published Studies

The summaries provided herein emphasize (a) dose per colony, (b) season of fumagillin application, and (c) the time elapsed between the fumagillin application and the observed outcome, which are factors that most authors report. However, every experiment differs, and it is not possible to describe the complexities of each study within the scope of this review. In numerous cases, fumagillin was applied multiple times. If the applications occurred within a single season (such as every week for four weeks), and measurements were taken subsequently, the sum of the treatments is reported as the dose. If treatments were applied in different seasons (such as spring and fall) with outcomes measured between treatments, multiple dates and doses are reported. In experiments where fumagillin was applied at multiple doses with a single control group, or where fumagillin was crossed with some other treatment, each treatment group is compared to the most similar group that did not receive fumagillin. If the same variable was measured on multiple dates, a significant difference is reported if any date was significantly different from the control; otherwise, the largest measured difference from the control is reported in the summary.

Many authors did not report tabulated measurements for each group in their studies. Some reported only the statistical significance (or lack thereof) of fumagillin, averaged across other treatments; others (particularly before 1980) presented no statistical comparisons. Occasionally, there is detailed information for some subgroups but not others. Where necessary and possible, effects have been estimated from figures or calculated from the measured values that have been reported.

## 3. Results

Fifty publications met the criteria as controlled field trials of fumagillin. Nevertheless, these articles probably do not represent an exhaustive list of studies that would meet the search criteria. Some papers that appeared likely to describe field studies could not be obtained. Inaccessible articles were disproportionately older, published in obscure trade journals or conference abstracts, and in languages other than English. In addition, some trials that met the search criteria were not described as trials of fumagillin by the authors (e.g., [20]) and could easily have been overlooked. Furthermore, some relevant and well-regarded studies did not meet our criteria because the treated and untreated colonies belonged to different beekeepers (e.g. [21]). One study [22] that did not include an equivalent untreated control group was accepted. It was a dose–response trial, and the linear regression partially compensates for the lack of a control group.

Twenty-two of the papers summarized herein were published before 1980, and twenty have been published since 2005, which is approximately the date when *N. ceranae* was recognized as a concern in European honeybees [7]. Only eight studies were found from the twenty-five-year period between 1980 and 2004. All the studies published after 1955 appear to have used commercial formulations with the dicyclohexylamine salt of fumagillin.

### 3.1. Effects of Fumagillin on Nosema

#### 3.1.1. 1953–1979

The twenty-two publications prior to 1980 describe twenty-nine controlled trials that measured the effect of fumagillin on *Nosema apis* (Table 1). Twenty-five of these trials were conducted in North America, three in Europe, and one in Australia. Every trial reported reductions in nosema prevalence (the proportion of bees infected) or intensity (the average number of spores per bee) in at least some treated groups. During this period, only four papers provided statistical comparisons [23,24,25,26]. All four found that reductions in nosema prevalence were statistically significant (Table 1). Reductions in nosema were detected more consistently following treatment in the spring rather than in other seasons (Table 1: 46 of 48 spring-treated groups versus 22 of 30 groups without spring treatment). In addition, reductions in nosema appeared to be detected more frequently when the nosema measurement was taken shortly after the fumagillin treatment, and increases in nosema were sometimes found when intensity rather than prevalence was measured (Appendix A). The presentation of data in Table 1 is constrained to a single line for each experiment. However, when measurements taken within six weeks of treatment were viewed separately, reductions in nosema were reported in 89% of the treated subgroups (41 of 46) but in only 68% of treated subgroups (26 of 38) when measured after longer time periods. In the latter presentation of data, reductions in nosema prevalence (86%; 24 of the 28 treated groups) were found more consistently than reductions in intensity (77%; 43 of the 56 treated groups).

Nearly all of the spring experiments were conducted using recently hived package bees, and, in such cases, nosema was measured shortly after treatment. Doses as low as 35 mg per package colony were at least partly effective, but no dose or treatment schedule was shown to eradicate the infection. Girardeau [26] showed that approximately 68 mg provided near complete nosema control in queen mating nucs, while lesser amounts did not.

Only one report prior to 1980 [43] described nosema levels after applying fumagillin in the spring to overwintered colonies. Instead, colonies were given a large dose in the fall to inhibit a spring infection. Doses applied in the fall varied between 50 and 880 milligrams (mg) per colony, and nosema levels were tested many months after the treatment. Notably, 200 mg of fumagillin per colony came to be recommended as a fall dose [23,38,45], but several reports [23,24,34] already showed that this amount did not always fully control nosema. Doses above 350 mg/colony appear to have been more effective [34].

#### 3.1.2. 1980–2004

In studies published between 1980 and 2004, the nosema species was assumed to be *N. apis*, but since subsequent work has shown that *N. ceranae* was already present in many places [46], this identification should be taken with some caution. Only a few studies about fumagillin were published between 1980 and 2004 [22,47,48,49,50,51], but those reports were more robust than the earlier period in two respects: Most included larger numbers of replicates than in the earlier period, and all studies employed statistical comparisons. Each of the six studies that measured nosema revealed significant benefits associated with fumagillin (Table 2). Several studies from this period provide information about the dose of fumagillin required to prevent serious nosema infections in wintered colonies. Szabo and Heikel [22] showed that fumagillin doses of 400 or 600 mg per colony in the fall provide better control of nosema in the spring than the typical 200 mg dose. They also showed that a 42 mg spring dose following a 200 mg fall dose reduced nosema levels compared to the fall dose alone, but infections were still detectable [50]. Wyborn and McCutcheon [47] obtained near complete control with a similar approach but using three spring applications, each containing 100 mg fumagillin. Goodman et al. [48] reported that a spring dose of 100 mg per colony reduced spring spore counts by approximately half in two independent trials (one of which also incorporated a fall treatment).

#### 3.1.3. 2005–2023

Since 2005, 19 reports, describing 24 field trials, have shown the effect of fumagillin on nosema (Table 3). *Nosema ceranae* predominated in all studies where species was determined. None of the studies reported treatments applied to newly hived package bees. Just over half of the trials reported the effects of a single fall or winter treatment; eight involved twice-yearly applications, and one described four applications in a single year. Studies in this period have continued to be larger than the ones before 1980, but they often also had complex designs with crossed factors. In contrast to earlier periods, many of the studies have been conducted by authors who were seeking alternatives to fumagillin. Nevertheless, all but two reports showed lower nosema levels in at least some fumagillin-treated groups; only three showed increased nosema levels compared to control in any fumagillin-treated group, and only one study reported a statistically significant increase.

Doses applied in the fall ranged from 48 mg to 200 mg/colony, and doses applied in the spring ranged from 68 to 120 mg/colony. Both ranges are low compared to values that were historically required to attain full control of *Nosema apis*, particularly in full-sized colonies. The only research group that tested fall or winter doses above 200 mg [58,62] demonstrated better nosema control at the higher dose (Table 3). Nevertheless, even the low-dose trials consistently found reduced nosema prevalence and spore counts in the treated groups, especially when the measurements were taken within six weeks of the treatment (Table 3). As in earlier periods, all studies that measured nosema prevalence found positive treatment effects, while those that measured spore counts were somewhat variable (Appendix A).

### 3.2. Effects of Fumagillin Treatment on Colony Characteristics

#### 3.2.1. 1953–1979

Thirteen field trials published before 1980 reported the effects of fumagillin on colony survival, size, or productivity (Table 4). As mentioned above, most field trials during this period either involved spring treatments on package colonies or fall treatments on colonies intended for overwintering. Fumagillin was not applied to overwintered colonies in the spring except for one trial [29]. Most trials that measured survival, colony size, or honey production involved fall-treated colonies, and none provided statistical comparisons of the results.

The effect of fumagillin on colonies was clearly positive (Table 4; Appendix A) but not as consistent as the effect on nosema. Only one report described any fumagillin-treated groups that underperformed compared with the control group [36]. Three reports demonstrated increased colony survival in fumagillin-treated groups, and three found larger adult bee populations. All five trials that tested the effects of fumagillin on brood reported larger brood areas in some treated groups, and six of seven trials reported that fumagillin treatment led to greater honey production.

The four experiments involving newly hived package colonies reported increases in adult bee population, brood population, and honey production compared to control groups, except for the low (50 mg) doses in one study [36]. Only one study demonstrated benefits from a spring dose below 100 mg/colony [29].

None of the nine trials involving fall treatments reported that fumagillin harmed colonies in any way. Three trials reported improvements in colony survival; one reported larger adult bee populations in the spring (but the treated groups included a second antibiotic, oxytetracycline, that was not in the control; [41]); and one reported larger brood populations. Three of four trials reported that the fumagillin-treated colonies produced more honey than the controls, sometimes substantially more. In one study, fall doses in the 440-880 mg/colony range led to greater honey production than a 220 mg/colony dose [38].

#### 3.2.2. 1980–2004

Between 1980 and 2004, the strongest results of fumagillin treatment were found by Woyke [69], who applied a rather low dose of fumagillin to nucleus colonies in late spring and detected significant increases in adult bees, brood, and honey yield a few weeks later (Table 5). Similarly, Goodman et al. [48] found large increases in honey production following spring or fall and spring fumagillin treatments. Conversely, two large studies [22,46] failed to detect significant effects on colony survival or adult bee populations. However, these two results may not indicate that fumagillin was unhelpful. In one case, the dose was too low (42 mg/colony) to achieve full control of nosema [50], while in the other, spring colony populations were 2–2.5 frames greater in colonies treated with 300–600 mg fumagillin in the fall than at the lowest dose (200 mg) [22].

#### 3.2.3. 2005–2023

The discovery of *Nosema ceranae* in European honeybees has led to a considerable increase in the number of studies of fumagillin, and the effects of fumagillin on colony survival and productivity have been a major focus. Fourteen of twenty-one trials described the effects of a single application in the fall or winter. The length of follow-up for such trials was between five weeks and half a year. The remaining six trials studied the effects of treatment in the spring and fall, with an additional study examining the effects of four treatments per year; periods of follow-up in these studies were between 42 weeks and 3 years. No statistically significant negative effects of fumagillin were detected. One trial [56] reported significantly lower bee populations in some fumagillin-treated groups; however, the initial infections were, by design, almost an order of magnitude greater than the controls. Only the treated groups with initial infections similar to the control groups are included in the summary tables.

While results vary (Table 6; Appendix A), positive results are clearly more common than negative ones. At least seven publications (describing nine trials) demonstrated that fumagillin provided statistically significant benefits to colonies, while no author demonstrated significant harm. Four of thirteen trials reported significant improvements in colony survival; six of seventeen reported significant increases in adult bee populations; three of thirteen found significant increases in brood populations, and three of eight found significant increases in honey production. Six of the ten reports that failed to detect any statistically significant benefits to fumagillin treatment either applied low doses of fumagillin or included only a few fumagillin-treated colonies. In the remaining reports [55,57,62,68], it is unclear why there was no significant benefit to fumagillin treatment. The simplest explanation is that nosema may not have been the main limiting factor in their colonies.

## 4. Discussion

### 4.1. Effect of Fumagillin Dicyclohexylamine on Nosema

It is widely recognized that the chemical controls of pests and diseases tend to decline in effectiveness because organisms that develop mutations to tolerate the treatment gain a selective advantage [72]. Fumagillin has been used in honeybee colonies for approximately seventy years, and some authors have reasonably suggested that *Nosema apis* might have developed resistance or that fumagillin use might have encouraged the spread of *Nosema ceranae* [16,21]. However, such claims require evidence.

The clearest evidence for resistance would be a difference between the historic and recent measurements of concentrations of fumagillin needed to inhibit nosema infections using standardized assays; however, these have not been developed against *N. apis* and *N. ceranae*. One group applied varying concentrations of fumagillin to infected bees and measured the subsequent spore counts, but the treatments were mostly well below the effective range, and in any case, there are no comparable historic measurements [73]. Therefore, field reports may represent the best available data. Controlled field experiments clearly show that fumagillin reduced nosema prevalence and spore counts in the past (Table 1 and Table 2) and still does so today (Table 3) at similar doses.

Numerous authors [5,10,12] have suggested that fumagillin may be ineffective against *N. ceranae* infections or may actually increase spore counts. The only evidence in favour of this claim is a single lab-based study, where *N. ceranae*-infected bees that were exposed to extremely low levels of fumagillin contained more spores than untreated bees [73]. Cage-based experiments play an important role in honeybee science because they can allow for more sensitive measurements of phenomena that are difficult to detect in the field. However, the extrapolation of lab studies to the field is rarely straightforward. Since the time *Nosema ceranae* became widespread, at least 24 controlled field trials have measured within-colony nosema levels following fumagillin treatment (Table 3). All but one showed either significantly positive or nonsignificant effects. Taken together, colony-level results clearly show that fumagillin DCH treatments reduce the prevalence and intensity of *N. ceranae* infections. Claims to the contrary are not well supported.

There is some evidence from cage trials that the concentration of fumagillin required to inhibit *N. ceranae* may be higher than for *N. apis* [73]. Fumagillin DCH controls both species of nosema but might not affect them equally [54]. None of the field trials identified here directly compared the effect of fumagillin against the two species, and there are too many differences among trials to draw inferences by comparing historic reports about *N. apis* against recent ones involving predominantly *N. ceranae*.

### 4.2. Effects of Fumagillin Dicyclohexylamine on Honeybee Colonies

The studies reviewed in this report do not support the claim that fumagillin DCH harms honeybee colonies, because not a single field trial showed statistically significant harm to colonies following exposure to the drug (Table 6). Furthermore, studies published since 2005 (Table 6) are similar to earlier studies (Table 4 and Table 5; also see Appendix A) in showing a mix of positive and neutral effects on colony-level characteristics. Thus, there is no evidence that the value of fumagillin to the colony has declined since the advent of *N. ceranae*. It is true that in cage trials, fumagillin DCH shortens the life of uninfected bees [74,75], but this effect is not apparent at the colony level. It seems that the benefits of reducing nosema infections, even at a low level, outweigh any harm to the colony.

The effect of fumagillin on the survival, size, and productivity of the honeybee colony has always been less consistent than its effect on nosema. A visual comparison of Table 1, Table 2 and Table 3 against Table 4, Table 5 and Table 6 shows fewer dark green (significant improvement) effects and more yellow (similar to control) effects in the latter. This is not surprising. Fumagillin’s only function in the hive is to inhibit nosema; it is the increase in the number of healthy bees that produces colony-level benefits. If nosema is not a major growth-limiting factor, fumagillin will provide little benefit.

### 4.3. Economic Thresholds

Some authors [65,76,77] describe an economic threshold for nosema infections, usually one million spores per bee. This number appears to be derived from an unpublished and inaccessible work by Jaycox (1980; as cited in [77]), although a similar statement can be found in at least one earlier source [78]. Since fumagillin trials that report economic effects usually also report measurements of nosema, they should shed light on the conditions at which fumagillin treatment becomes worthwhile. For this reason, the prevalence or intensity of infection in the untreated groups is reported in Table 4, Table 5 and Table 6. Unfortunately, the attempt to make meaningful comparisons across studies proved impossible, because bees were sampled from different locations in the hives, and nosema was evaluated in different ways. Experiments were also conducted in different seasons and regions, with differently managed colonies. Consequently, controlled field trials do not provide evidence in favour of any particular threshold value. When one million spores per bee is the level used as a threshold, colonies above this threshold are less productive on average than those below [79], but the level itself is arbitrary.

Inconsistency in the method of reporting infections is a particular barrier to the development of an evidence-based economic threshold. Early authors from the time of White [80] monitored the proportion of infected bees, but others used spore counts from pooled samples [81], which is less laborious, and this approach became more common. However, infection prevalence predicts honey yield better than spore counts [82]. Fumagillin trials that used a prevalence-based method show that fumagillin lowers nosema prevalence. Spore-count-based trials have mostly agreed with this result, but some have found neutral effects or insignificant increases (Appendix A). The reason for the discrepancy is easy to understand. Spore counts in individual bees vary between zero and at least tens of millions [83]. Given such large differences among bees, the sickest bee in a pooled sample is likely to contribute to the vast majority of the spores. Consequently, a single pooled sample mainly measures an individual-level effect, and its relationship to the colony-level infection is only probabilistic. The prevalence method is more appropriate because it measures the proportion of bees that are not impaired by the disease, which directly affects the colony’s productive capacity.

Consider the case where pooled samples of 100 bees per colony are taken, and 80% of samples are nosema-positive. If one makes the simplified assumption that there are no important differences in nosema prevalence among colonies, the measurements can be regarded as random samples from a single population, and the binomial theorem can be applied to estimate the proportion of infected individual bees [84]. Since eighty percent of the samples were positive, twenty percent were nosema-free. That is, *p*^n^ = 0.2, where n is the number of bees in each sample (=100), and p is the proportion of uninfected individual bees. Here, *p* = 0.984, which implies that only 1.6% of bees were infected with nosema. When 100-bee samples are taken from a population with an infection rate of 1.6%, approximately 92% of samples will contain three or fewer infected bees. The spore counts from such samples represent the colony poorly. This problem is likely to be most severe when the nosema prevalence is lowest (e.g., samples from the brood nest, sample dates in summer, etc.). Ironically, it may also be worse when the number of bees in each sample is large because the proportion of nosema-free samples (which reflect infection prevalence alone) decreases, while the proportion of samples with a few sick bees increases.

There is an additional problem with current recommendations based on threshold values: It is not obvious how samples from a small number of colonies should affect treatment decisions applied to larger groups. From experience, if fifteen or twenty samples are taken per apiary, some of them may contain fewer than one million spores per bee, even during the spring peak of infection in untreated colonies; similarly, some are above 1 million spores per bee, even if tested in late summer or immediately after fumagillin treatment has been completed. Thus, the effort spent on testing does not result in clear guidance. An appropriate standard should specify the number of tests, the proportion of test results that may exceed the threshold, and, for large operations, the number of apiary locations to be tested.

### 4.4. Nosema Management and Dose

As Bailey [24] recognized, control of nosema involves addressing three sources of infection: (1) the active infection in individual bees, (2) spores in the hive, and (3) environmental exposure. It was also recognized early that fumagillin did not kill dormant spores [2] and therefore could not be used as a disinfectant. Therefore, control of nosema requires the approach taken by Bailey: movement restrictions and quarantine to limit environmental exposure, replacing contaminated comb with new or disinfected comb, and antibiotic treatment. Only a few of the studies examined [39,40] investigated the use of fumagillin together with a system of disinfection and quarantine. Unless the spores are removed, control of nosema will depend on long-term and fairly high doses of antibiotic, whether or not that antibiotic is fumagillin.

The optimum timing of fumagillin application is unclear. Most trials of fumagillin have been conducted in temperate regions with distinct spring and fall periods and clear seasonal patterns of nosema infection. Positive effects have been detected more frequently in studies that included a spring treatment, but this is not conclusive. Before 1980, spring treatments were mainly used with package bees or other special situations, such as queen mating nucs. Since 1980, most studies have included a fall or winter treatment, and some have also included a spring treatment. Studies with multiple treatment periods have usually reported colony-level benefits (Table 4, Table 5 and Table 6).

The recommended spring dose of fumagillin (100 mg fumagillin base) was developed for the North American package bee industry. Beekeepers in northern regions killed their hives in the fall when nosema infection levels were lower than those developing during confinement over the winter. In the spring, packaged bees were shipped north and hived on a comparatively spore-free comb. As a result, the bees themselves were the main source of infection. A three- or four-week treatment period with a small amount of fumagillin suppressed nosema development before new cohorts of nosema-free bees replaced the existing adult population.

Wintered colonies, even when treated in the spring, are not the same as packages because they are normally larger and nearly always housed in the same hive equipment as during the winter, which has become fouled because of ongoing infections. If the beekeeper wants to control nosema in wintered colonies without extensively manipulating and disinfecting them, the antibiotic must be applied until the risk of transmission naturally declines, which implies a considerably longer treatment period and larger dose. Spring applications of 42 mg or 100 mg led to average spore counts well above one million per bee [46,48]. Wyborn and McCutcheon [47] obtained better control by applying 100 mg three times at two-week intervals.

The classical approach for wintered colonies was to apply a single large dose (often 200 mg) in the fall instead of a spring treatment. However, the cost of treatment was a major consideration at that time [35,38,85], and optimal honey production appeared to follow a higher dose, 440 mg [38]. Fall doses in the range of 400 to 600 mg per colony resulted in less nosema the following spring, compared to the 200 mg dose [22,24,34]. The complete control of nosema in over-wintered colonies by means of fumagillin alone has probably always required more than the 200 mg/colony fall treatment or 100 mg/colony spring treatment.

### 4.5. Toxicity Concerns

Not all of the questions that have been raised about fumagillin use relate to its effect on nosema or its economic value to beekeepers. Concerns about the effects of antibiotic residues in food or the environment are widespread. These questions are beyond the scope of this review, but some factors seem apparent: (1) dicyclohexylamine, but probably not fumagillin, has some negative effects on healthy bees and should possibly be replaced [75]; (2) fumagillin dicyclohexylamine at very high doses has negative effects on mammals including humans [2]; and (3) fumagillin and dicyclohexylamine are sometimes detected in honey but at low concentrations [4,86]. The dicyclohexylamine counter-ion appears to be both more persistent in hive products and more toxic to mammals than fumagillin itself and may be a cause of concern [3]. However, the current formulation of fumagillin has been widely used in commercial apiaries for about seventy years, and we found no cases of harm to humans or bees cited in the fifty publications reviewed.

## 5. Conclusions

Fumagillin dicyclohexylamine inhibits both species of nosema that commonly infect the honeybee colony, and its effectiveness in the field has not perceptibly declined after seventy years of widespread use. Treatment may shorten the lifespan of otherwise healthy individual bees in cages, but at a colony level, fumagillin clearly provides a net benefit. None of the studies we reviewed reported harm to honeybee colonies from fumagillin use, and statistically significant benefits were regularly reported. However, details of dose, timing, and treatment threshold have not been optimized, and more work would be appropriate in these areas. An appropriate strategy would be to establish a clear relationship between a particular measure of nosema (for example, prevalence among older bees at the end of winter) and a particular colony-level outcome (for example, honey production during the following season). Subsequently, treatment strategies could be compared based on their ability to disrupt this relationship. In addition, there is very little information about the effects of treatment on queen survival, and studies that report effects on colony survival have usually been underpowered. Finally, beekeepers can expect to achieve better nosema control if they use hive disinfection procedures and fumagillin together.

## Figures and Tables

**Table 1 insects-15-00029-t001:** Studies measuring the effects of fumagillin treatment on *N. apis* infections in *A. mellifera* colonies (1953–1979).

Year	Reference	Trial Size ^a^ (Colonies)	Treatment ^b^ (Season; Dose)	Delay ^c^ (Weeks)	Effect ^d,e^			
					Intensity		Prevalence	
1953	[23]	41/80	F; 200	20			1	
1953	[27]	13/27	Sp; 190	3			1	
1954	[28]	1170/2340	Sp; 172	3			1	
1955	[24]	29/64	F; 50–500	26			1	3
1955	[29]	52/78	Sp; 35–70	3			4	
		36/72	F; 140	5			2	
1957	[30]	1/22	Sp; 100	6	1			
1957	[31]	10/12	Sp; 10–170	2	2	3		
		26/55	F; 200	20	1	2		
1957	[32] ^f^	28/56	Sp; NS	3	1			
		50/75	F; 75 or 100	6	2			
1957	[25] ^g^	35/119	F; 216	26			1	
1961	[33]	24/36	Sp; 150	6			4	
1962	[34]	3/8	F; 176–706	6–33	3			
1962	[35]	57/81	Sp; 147	3			2	
1969	[36]	16/24	F; 100 or 200	1–39	2	2		
		16/24	Sp; 50 or 100	0–6	4			
1969	[37]	8/18	F; 200	2–35	2			
		16/36	Sp; 50	2–8	4			
1970	[38]	12/48	F; 220–880	2–43				
1970	[39] ^h^	100/200	F & Sp; NS	0–10	2			
1972	[26] ^i^	56/112	Sp; 68	0–6			1	
		80/120	Sp; 17 or 34	0–6			1	1
1972	[40] ^h^	40/100	W; 200	0–25	1	1		
1973	[41]	150/200	F, W, or F&W; 200	0–35			1	2
		NS	F&W; 100 or 200	0–35			2	
1973	[42] ^j^	40/50	Sp; 100–300	3–11	4			
1977	[43]	6/12	Sp; 100	1–8	2			
1979	[44]	5/10	Sp; 100	0–10	1		1	

^a^ Fumagillin-treated colonies/total colonies. ^b^ Seasons indicated as F: fall; W: winter; Sp: spring; Su: summer. Doses are in milligrams per colony per season. NS: not stated in the original report; ^c^ approximate time in weeks between the last indicated date of fumagillin treatment and the date of nosema measurement. Where a range of dates is shown, repeated measurements were taken. ^d^ Colour codes: Pink: the fumagillin-treated group had more nosema than control (not statistically significant or significance not reported); Yellow: nosema level in the fumagillin-treated group was within 10% of control and not statistically different (or significance not reported); Light Green: the fumagillin-treated group had less nosema than control (not statistically significant or significance not reported); Dark Green: the fumagillin-treated group had significantly less nosema than control. See Methods for details. ^e^ Many studies included multiple fumagillin-treated groups. Numbers show how many fumagillin-treated groups gave the response indicated by the colour code. If results were reported for fumagillin as a factor, but not for the individual groups, no numbers are shown. ^f^ Results were given as percent infection, but in fact, they were estimated from a spore count. ^g^ Unusual significance threshold: *p* = 0.06 was considered significant. The dose is shown as stated but appears incorrect: 10 L syrup at 0.5 g Fumadil-B/L would be 100 mg fumagillin per colony (Fumadil-B contains 20 mg fumagillin per gram). ^h^ The study included treatments applied during the production of package bees and after hiving. Only the after-hiving results are summarized. ^i^ The “colonies” were queen mating nucs. ^j^ Fumagillin treatments applied in grease patties were described as ineffective by the original author, but nosema was still notably lower than control in these groups.

**Table 2 insects-15-00029-t002:** Studies measuring the effects of fumagillin treatment on presumed *N. apis* infections with possible covert infections of *N. ceranae*, in *A. mellifera* colonies (1980–2004).

Year	Reference	Trial Size ^a^ (Colonies)	Treatment ^b^ (Season; Dose)	Delay ^c^ (Weeks)	Effect ^d,e^	
					Intensity	Prevalence
1981	[49]	12/24	Sp; 200 or 400	0–9	3	
1986	[50]	60/128	Sp; 68	1, 3	1	
1987	[22] ^f^	194/194	F; 200–800	0, 25		
1987	[47]	20/30	Sp; 300	0–3	2	
1990	[48]	60/120	Sp; 100	NS	1	
		100/200	F; 100 & Sp; 100	NS	1	
1996	[51] ^g^	19/44	Sp; NS	4		2

^a^ Fumagillin-treated colonies/total colonies. ^b^ Seasons indicated as F: fall; W: winter; Sp: spring; Su: summer. Doses are in milligrams per colony per season. NS: not stated in the original report; ^c^ approximate time in weeks between the last indicated date of fumagillin treatment and the date of nosema measurement. Where a range of dates is shown, repeated measurements were taken. ^d^ Colour codes: Light Green: the fumagillin-treated group had less nosema than control (not statistically significant or significance not reported); Dark Green: the fumagillin-treated group had significantly less nosema than control. See Methods for details. ^e^ Many studies included multiple fumagillin-treated groups. Numbers show how many fumagillin-treated groups gave the response indicated by the colour code. If results were reported for fumagillin as a factor, but not for the individual groups, no numbers are shown. ^f^ No untreated group. The expected effect was a dose–response, not a difference from the control. ^g^ Concentration, not dose, was standardized: 1.3 g fumagillin DCH in 1 kg honey-sugar patties. Assuming the commercial product Fumadil-B was used, this would be 26 mg of fumagillin per kg.

**Table 3 insects-15-00029-t003:** Studies measuring the effect of fumagillin treatment on presumed or confirmed *N. ceranae* infections, or mixed *N. apis* and *N. ceranae* infections, in *A. mellifera* colonies (2005–2023).

Year	Reference	Trial Size ^a^	Treatment ^b^ (Season; Dose)	Elapsed Time ^c^	Nosema Species ^d^		Effect ^e,f^					
					Stated	Tested	Intensity				Prevalence	
2005	[52] ^g^	55/162	Sp & F; 80	0–26	a	No	1					
2008	[53]	05/10	F; 120	0–22	c	Yes					1	
		18/50	F; 120	2–43	c	Yes					1	
2011	[54]	48/70	F; 120,160	2–56	c	Yes					4	
2011	[55]	26/47	F; 190	26	c	Yes	1					
		52/105	F; 190	26	c	Yes	5		2			
2011	[56] ^h^	40/48	F; 102, 120	5	c	Yes	5					
2012	[57]	96/190	W; 200	13	c	No	2	1		1		
2013	[58]	21/42	W; 200 or 400	7	c	No	1		1			
2013	[59] ^i^	30/50	F, W, Sp & Su; 120	<6 >6	m m	Yes Yes					3	
											1	2
2015	[60]	10/50	F; 200	30	c	No	1					
2016	[61] ^j^	32/48	F; 102	<4	c	Yes	2		2			
2017	[62]	44/72	W; 120 or 360	1 8	c c	Yes Yes	4					
							2	1		1		
2018	[63] ^k^	4/16	F, Sp, & Su; 100	26	c	Yes	1					
2018	[20] ^l^	80/160	Sp, F; 120	< 6 6–26	NS NS	No No						

2019	[64]	6/62	F & Sp; 120	1 20	c c	Yes Yes					1	
											1	
2020	[65] ^m^	36/72	Sp; 100 & F; 200	<6 6–23	c c	Yes Yes	1					
							1					
2021	[66]	6/66	Sp; 100 & F; 200	3 30	c c	Yes Yes	1					
							1					
2021	[67]^l^	96/127	Sp; 68 & F; 120	< 6 32	c c	Yes Yes						
							2		1			
		96/126	Sp; 68 & F; 48	< 6 32	m m	Yes Yes						

2021	[6] ^l^	18/56	NS; 101	0	c	Yes	1					
2023	[68] ^n^	30/90	W; NS	22	NS	No	1					
		12/30	W; 100,200	7	NS	No	1					
		12/30	W; 100,200	8	NS	No	1					

^a^ Fumagillin-treated colonies/total colonies. ^b^ Seasons indicated as F: fall; W: winter; Sp: spring; Su: summer. Doses are in milligrams per colony per season. NS: not stated in the original report; ^c^ approximate time in weeks between the last indicated date of fumagillin treatment and the date of observation. Where a range of dates is shown, repeated measurements were taken; results usually varied among dates and only the largest observed effect is shown for each group. ^d^ Species. a: *N. apis*; c: *N. ceranae*; m (mixed): both *N. apis* and *N. ceranae* were present; NS: not stated by original authors. ^e^ Colour codes: Red: the fumagillin-treated group had significantly more nosema than control; Pink: the fumagillin-treated group had more nosema than control (not statistically significant or significance not reported); Yellow: nosema level in the fumagillin-treated group was within 10% of control and not statistically different (or significance not reported); Light Green: the fumagillin-treated group had less nosema than control (not statistically significant or significance not reported); Dark Green: the fumagillin-treated group had significantly less nosema than control. See Methods for details. ^f^ Many studies included multiple fumagillin-treated groups. Numbers show how many fumagillin-treated groups gave the response indicated by the colour code. If results were reported for fumagillin as a factor, but not for the individual groups, no numbers are shown. ^g^ The following details are unclear: whether the nosema measurements occurred before or after the spring fumagillin treatments and whether all colonies in the fumagillin group were treated or only those with more than 2 million spores per bee. A significant statistical comparison was provided for the fumagillin treatment, but it appears to be the comparison of one year against another, instead of against the control. ^h^ Three of the fumagillin-treated groups had an initial spore count ten times that of the control. ^i^ Treatments were applied in one, two, or four seasons; the time between treatment and measurement varied by group. ^j^ Treatments and measurements were both applied monthly through the fall and winter; the dose indicated is the dose per month. In one group, colonies were treated only if the spore count was >350 000 spores/bee. ^k^ The authors regarded the fumagillin treatment as unsuccessful. However, colonies in the fumagillin-treated group were essentially nosema-free in spring, which is the intended effect of the fall treatment. The effectiveness of the spring and summer treatments is unclear and not shown. ^l^ Factorial experiment: model effect is shown instead of group-by-group comparisons. ^m^ The significant differences were observed only following the spring treatment. ^n^ “Label” treatments were used without specifying exact quantities. For the first trial, it was not obvious whether the treatment was a spring dose, fall dose, or other quantity at the label concentration.

**Table 4 insects-15-00029-t004:** Studies measuring the effects of fumagillin on survival, size, or productivity of *Apis mellifera* colonies infected with *N. apis* (1953–1979).

Year	Reference	Trial Size ^a^	Prior Infection ^b^	Treatment (Season; Dose) ^c^	Total Time (Weeks) ^d^	Effects ^e,f^			
1955	[29]	52/78	25%	Sp; 35–70	9	Brood (4)			
					13 13	Adults (4)			
						Honey (4)			
		26/65	NS	F; 140	26 26	Survival (2)			
						Adults (1)			
		36/72	31, 90%	F; 140 & Sp; NS	18	Survival (1)		Survival (1)	
					43	Honey (1)			
1957	[30]	1/22	0%	Sp; 200	6	Brood (1)			
1957	[25]	35/119	4%	F; 216	26	Survival (1)			
1962	[35]	57/81	10, 50%	Sp; 147	3	Brood (2)			
					26	Honey (2)			
1969	[36]	16/24	2 msb	F; 100 or 200	78	Survival (1) ^g^			
					43	Honey (3)			
		16/24	22 msb	Sp; 50 or 100	7	Brood (1)	Brood (1)		Brood (2)
					7	Adults (1)		Adults (3)	
					18	Honey (1)	Honey (2)		Honey (1)
1969	[37]	8/18	NA ^h^	F; 200	35	Brood (1)		Brood (1)	
1970	[38]	12/48	NS	F; 220–880	26	Survival (1) ^g^			
					43	Honey (6)			
1972	[40]	40/100	0.01 msb	W; 200	39	Honey (2)			
1973	[41] ^i^	150/200	NS	F, W, F&W; 200	21	Adults (3)			
		NS/NS	NS	F&W; 100 or 200	21	Adults (2)			

^a^ Fumagillin-treated colonies/total colonies; ^b^ measured infection in the control group at the time of fumagillin treatment; msb: million spores per bee, %: percent infected. ^c^ Seasons indicated as F: fall; W: winter; Sp: spring; Su: summer. Doses are in milligrams per colony per season. NS: not stated in the original report; ^d^ approximate time in weeks between the last indicated date of fumagillin treatment and the date of observation. Where a range of dates is shown, repeated measurements were taken; results usually varied among dates and only the largest observed effect is shown for each group. ^e^ Colours: Pink: the fumagillin-treated group was worse than control (not statistically significant or significance not reported); Yellow: the fumagillin-treated group was within 10% of control and not statistically different (or significance not reported); Light Green: the fumagillin-treated group was better than control (not statistically significant or significance not reported); Dark Green: the fumagillin-treated group was significantly better than control. See Methods for details. ^f^ Many studies included multiple fumagillin-treated groups. Numbers show how many fumagillin-treated groups gave the response indicated by the colour. ^g^ Because of the small number of colonies in each subgroup, groups are combined. ^h^ NA: not applicable. Prior infection was measured, but a large inoculum was added, so the initial value does not represent the severity of the infection. ^i^ The treated groups, but not the control, were also administered oxytetracycline.

**Table 5 insects-15-00029-t005:** Studies measuring the effects of fumagillin treatment on survival, size, or productivity of *A. mellifera* colonies believed to be infected with *N. apis*, with possible covert infections of *N. ceranae* (1980–2004).

Year	Reference	Trial Size ^a^	Prior Infection ^b^	Treatment (Season; Dose) ^c^	Total Time (weeks) ^d^	Effects ^e,f^
1981	[49] ^g^	4/24	6 msb	Sp; 200	6	Brood (1)
1984	[69]	12/24	NS	Sp; 71	6	Adults (1)
					6	Brood (1)
					8	Honey (1)
1986	[50]	60/126	15 msb	Sp; 42	4	Survival
1987	[22] ^h^	194/194	3 msb	F; 200–800	37	Survival
					37	Adults
1990	[48]	60/120	NS	Sp; 100	NS	Honey
		80/160	NS	F; 100, Sp; 100	NS	Honey
		100/200	NS	F; 100, Sp; 100	NS	Honey
1994	[70] ^i^	10/15	NS	F; 25	1	Queen Survival
					1	Uncapped Brood
					1	Capped Brood

^a^ Fumagillin-treated colonies/total colonies; ^b^ measured infection in the control group at the time of fumagillin treatment; msb: million spores per bee, %: percent infected. ^c^ Seasons indicated as F: fall; W: winter; Sp: spring; Su: summer. Doses are in milligrams per colony per season. NS: not stated in the original report; ^d^ approximate time in weeks between the last indicated date of fumagillin treatment and the date of observation. Where a range of dates is shown, repeated measurements were taken; results usually varied among dates and only the largest observed effect is shown for each group. ^e^ Colours: Pink: the fumagillin-treated group was worse than control (not statistically significant or significance not reported); Yellow: the fumagillin-treated group was within 10% of control and not statistically different (or significance not reported); Light Green: the fumagillin-treated group was better than control (not statistically significant or significance not reported); Dark Green: the fumagillin-treated group was significantly better than control. See Methods for details. ^f^ Many studies included multiple fumagillin-treated groups. Numbers show how many fumagillin-treated groups gave the response indicated by the colour. ^g^ Twelve colonies were treated with fumagillin, but eight of them (two treatment groups) also received other treatments with no direct control for the fumagillin portion of the treatment. ^h^ Dose–response trial with no untreated control; effect would have been indicated by a significant linear regression. ^i^ Nucleus colonies were used.

**Table 6 insects-15-00029-t006:** Studies measuring the effect of fumagillin treatments on survival, size, or productivity of *A. mellifera* colonies with presumed or confirmed *N. ceranae* infections, or mixed *N. apis* and *N. ceranae* infections (2005–2023).

Year	Reference	Trial Size ^a^	Prior Infection ^b^	Treatment (Season; Dose) ^c^	Total Time (Weeks) ^d^	Effects ^e,f^	
2005	[52]	55/162	5.1 msb	Sp; 80, F; 80	104 104 104	Brood	
						Adults	
						Honey	
2008	[53] ^g^	5/10	40.00%	F; 120	26	Survival	
					26	Adults	
					26	Brood	
		18/50	NS	F; 120	47	Survival	
2008	[71]	89/181	NS	W; 90	16 16	Adults (2)	
						Brood (2)	
2011	[54]	48/70	NS	F; 120 or 160	59	Survival (4)	
2011	[55]	26/47	1.3 msb	F; 190	28 28 28 28 28 28	Survival	
						Adults	
						Honey	
						Pollen	
						Capped Brood	
						Uncapped Brood	
		52/105	0.3–3.5 msb	F; 190	28	Survival	
2011	[56] ^i^	40/48	2 msb	F; 102, 120	5	Adults (2)	
2012	[57]	96/190	0.9, 1.4 msb	W; 200	13 13 13	Survival	
						Adults (4)	
						Brood (4)	
2013	[58]	21/42	0.5 msb	W; 200, 400	21 21	Survival	
						Adults (1)	Adults (1)
2013	[59] ^h^	30/50	29%, 56%	F, F & Sp, or FWSp & Su; 120	165	Survival	
					165	Queen Survival	
					60	Adults (3)	
					60	Brood (1)	Brood (2)
					48	Honey (1)	Honey (2)
2015	[60]	10/50	3.5 msb	F; 200		Adults	
2016	[61]	32/48	0.1, 0.9 msb	F; 102	5 5 5 5	Adults (4)	
						Brood (4)	
						Honey (4)	
						Pollen Stores (1)	Pollen Stores (1)
2017	[62]	44/72	2.3 msb	W; 120 or 360	12 12 12	Survival	
						Adults (1)	Adults (3)
						Brood (2)	Brood (2)
2018	[20]	80/160	6%	Sp & F; 120	44 44	Survival	
						Adults	
2019	[64]	6/62	8%	F & Sp; 120	48	Survival (1)	
					26	Adults (1)	
					21	Brood (1)	
					48	Honey (1)	
2021	[66]	06/66	3.9 msb	Sp; 100 & F; 200	9	Adults	
					9	Brood	
					15	Honey	
					52	Survival	
					52	Adults	
					52	Brood	
2021	[67]	96/127	3.5 msb	Sp; 68 & F; 120	45 45 45 45	Survival	
						Adults	
						Brood	
						Honey	
		96/126	2 msb	Sp; 68 & F; 48	42	Survival (1)	Survival (2)
					42	Adults	
					42	Brood	
					42	Honey	
2023	[68]	12/30	0.8 msb	W; 100,200	7	Adults	
					7	Brood	
		12/30	NS	W; 100,200	8	Adults	
					8	Brood	

^a^ Fumagillin-treated colonies/total colonies; ^b^ measured infection in the control group at the time of fumagillin treatment; msb: million spores per bee, %: percent infected. ^c^ Seasons indicated as F: fall; W: winter; Sp: spring; Su: summer. Doses are in milligrams per colony per season. NS: not stated in the original report; ^d^ approximate time in weeks between the first date of fumagillin treatment and the last date of observation. ^e^ Colours Pink: the fumagillin-treated group was worse than control (not statistically significant or significance not reported); Yellow: the fumagillin-treated group was within 10% of control; Light Green: the fumagillin-treated group was better than control (not statistically significant or significance not reported); Dark Green: the fumagillin-treated group was significantly better than control. See Methods for details. ^f^ Many studies included multiple fumagillin-treated groups. Numbers in parentheses show how many fumagillin-treated groups gave the response indicated by the colour. ^g^ Fumagillin-treated colonies reared significantly less brood in winter, but more in spring, than control colonies. The authors viewed winter brood rearing as a symptom of illness. In view of this and the clearly better subsequent result in the fumagillin-treated group, a negative effect of fumagillin on brood rearing is not shown. ^h^ Two control groups; the CS treatment (29% initial infection) is used for the comparisons. ^i^ Three treatment groups, representing 24 of the 40 fumagillin-treated colonies, had very high initial spore loads and no comparable control group. These results are not presented in the table.

## Data Availability

The data used in the paper are fully presented in Table 1, Table 2, Table 3, Table 4, Table 5 and Table 6 and associated references.

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
