# Peer review of "A Systematic Review of Fumagillin Field Trials for the Treatment of Nosema Disease in Honeybee Colonies"

_insects, 2024, doi:10.3390/insects15010029_

Round 1

Reviewer 1 Report

Comments and Suggestions for Authors

The authors review 50 carefully selected manuscripts of field studies in honey bees to investigate, whether fumagillin is still a useful and safe antibiotic to use against Nosema.

I find the topic interesting and valuable. However, the results of the review are correlative. Even though the authors invested a lot of work in identifying suitable manuscripts to include in their review, they did not go into the datasets to answer relevant questions using a meta analysis to create new graphs and they did not use statistical tests.

The authors conclude that the use of fumagillin seems safe.

This conclusion is not justified, I think. There were not statistical tests conducted to show this. Lab experiments that might have show otherwise were not included in the review. There was no mention of additive effects of fumagillin and parasites or fumagillin and pesticides, either.

358-363 "the effects of fumagillin was clearly postive".

In my opinion the authors did not test for the effects of fumagillin on colonies. They reviewed papers that stated that fumagillin treated colonies had certain phenotypes compared to untreated colonies. This does NOT mean that fumagillin per se has a positive effect on colonies.

In my eyes, the study would profit from asking the authors of the reviewed manuscripts (at least the ones after 1980) for original data files in order to conduct a meta analysis.

Please Note: Even if a study is written in a language other than English, it does not  mean it is "not accessible". You just need to find someone who speaks that other language.

Generally, I would have preferred for the data to have been presented in figures and more quantitatively  than just qualitatively in tables with the green, yellow red color code.

In the results section the data were often described as "seven out of 9 studies" etc. I would have preferred a figure showing the percentage of studies for this type of result, with an error bar.

 Why not create a figure with number of studies with a spring application - dose, then fall application - low dose and high dose - number of studies with positive results, number of studies with negative results and a standard error of the mean or some such? 

Questions that I had were:

What were the differences in effects of fumagillin when applied in spring versus applications in winter, what is the current recommendation based on this paper?

What are the minimally effective doses that are recommended to be used based on this paper?

Author Response

Comments: Reviewer One

General response to Reviewer One

Reviewer One has provided a considerable number of comments, many of which communicate the same message: that we should conduct a meta-analysis, rather than a systemic review. We feel these comments may reflect a lack of familiarity of the nature of systematic reviews, as well as the limitations of data availability in the studies being considered.

Specifically, the following comments were given:

However, the results of the review are correlative. Even though the authors invested a lot of work in identifying suitable manuscripts to include in their review, they did not go into the datasets to answer relevant questions using a meta analysis to create new graphs and they did not use statistical tests.

We have briefly addressed the distinction between a systematic review and a meta-analysis in the introduction to the document. More detailed information can be found at reference [18] and at https://training.cochrane.org/handbook/current/. Meta-analyses are a type of systematic review that uses statistical methods to combine results across studies. We agree that in many circumstances this can be highly desirable, such as when comparing results across several studies with low power and very similar protocols. The combined results of such studies can lead to a better estimate of effect size and statistical significance. However, when the differences in study protocols and circumstances are large and important, it is inappropriate to combine their results.

In our view, the diversity of methods and conditions used in fumagillin field trials over the last seventy years precludes a statistical approach. For example, what would be the appropriate way to combine the findings of a study which measured the proportion of sick foragers one week after a single fumagillin application in spring, with one that measured the average spore count of bees collected from inside the hive on multiple dates and following multiple treatments? Since each study differs from every other in multiple important (structural) ways, the attempt to compress our findings into an effect estimate and P value would either involve combining incompatible measurements (producing an uninterpretable result) or selecting a handful of relatively similar studies while arbitrarily excluding the majority. In contrast, our systematic review includes all studies that meet the screening criteria relevant to our research questions, and these have been succinctly summarized in the data tables in an informative manner.

The authors conclude that the use of fumagillin seems safe. This conclusion is not justified, I think. There were not statistical tests conducted to show this.

The request for statistical comparisons is another way of requesting a meta-analysis which, as explained above, would be an inappropriate way of summarizing this data. Furthermore, it is untrue that the data contain no statistical tests. When the original authors reported the results of a statistical comparison, we have summarized those results in the tables.

We feel it is not correct to say that the results of the review are correlative or that conclusions based on the totality of studies are unjustified without a statistical procedure. First, each study in the dataset was, by screening criterion, a randomized controlled trial with a clearly defined fumagillin treatment. Secondly, the presentation is not devoid of relevant statistics because some of the original authors conducted statistical tests, and we have included the results of those tests. Thirdly, we have also summarized non-significant results by clearly defined methods (except for one or two cases where they were unavailable), and so have not made the error of "vote counting based on statistical significance" (again from the Cochrane website). Fourthly, the function of statistical tests is to aid in evidence based conclusions. When the evidence is not ambiguous - for example when independent trials consistently report similar or compatible results - one does not require a calculation to identify the reasonable belief.

In my eyes, the study would profit from asking the authors of the reviewed manuscripts (at least the ones after 1980) for original data files in order to conduct a meta analysis.

Again, this is a request to replace the existing study with a totally different one that would not answer the questions presented in the introduction. The attempt to obtain original data files from dozens of publications would take many months, and is highly unlikely to yield enough of the original datasets. Of note, all of the principal investigators from the period before 1980, and half of those from the period between 1980 and 2005, are deceased.

Generally, I would have preferred for the data to have been presented in figures and more quantitatively than just qualitatively in tables with the green, yellow red color code.

The colour coding method which we have chosen is a version of "vote counting based on direction of effect" (Cochrane website cited above, chapter 12). The reviewer will notice, correctly, that this method is less powerful than a meta-analysis. However, it is also more broadly applicable. It can be applied appropriately when, as here, the effect of treatment is measured differently in different studies, or when it is not reasonable to expect that effect sizes would be similar across studies.

In the results section the data were often described as "seven out of 9 studies" etc. I would have preferred a figure showing the percentage of studies for this type of result, with an error bar.

Why not create a figure with number of studies with a spring application - dose, then fall application - low dose and high dose - number of studies with positive results, number of studies with negative results and a standard error of the mean or some such?

We have complied with this request with the addition of several supplemental figures (S1-S4).

S1 cited at line 184; S2 cited at line 332; S3 cited at line 365; and S4 cited at line 434; Figure captions at lines 665-682.

Though this provides a quicker visual way to compare the results of the many studies under review, a number of assumptions and generalizations needed to be made to discretely categorize the results to produce the figures. As such, we feel this is a less accurate way of presenting the data than those summarized in the tables and have thereby placed them in the supplemental section. The assumptions used to make the figures are listed in the captions.

Lab experiments that might have show otherwise were not included in the review.

It is essential to the conduct of a systematic review that the questions of interest, method, and scope be precisely defined. The reviewer is correct that lab trials were excluded from the review, and the reason is simple: our subject was the findings of field trials. It would not be a simple matter to evaluate lab trials and field trials of honey bees in the same review. Further, it is important to note that we have not ignored lab trials or contrary results. Several lab trials - precisely those whose findings may appear to conflict with the field trials [73-75] - are addressed prominently in the discussion (lines 504; 485; 494; 517; 640) .

Please Note: Even if a study is written in a language other than English, it does not mean it is "not accessible". You just need to find someone who speaks that other language.

We certainly agree with the reviewer that if a study is not in English that it does not mean that it is “not accessible”. We did seek translations of any relevant non-English language studies that could be located.

Please note that our review states the following:

“Inaccessible articles were disproportionately older, published in obscure trade journals or conference abstracts, and in languages other than English.”

This does not mean that we did not consider non-English language studies, only that of the studies we could not locate copies of, a higher proportion happened to be non-English. Many non-English papers met the screening criteria and were included; others were examined but did not meet the screening criteria. The statement simply expresses that we could not obtain every document that our search process suggested might be relevant.

We also point out that such statements are required as a part of the statement of limitations in systematic reviews, and does certainly not reflect any lack of effort to locate foreign-language publications on our part.

Other comments by Reviewer One:

The authors review 50 carefully selected manuscripts of field studies in honey bees to investigate, whether fumagillin is still a useful and safe antibiotic to use against Nosema.

I find the topic interesting and valuable.

We are glad to hear this.

There was no mention of additive effects of fumagillin and parasites or fumagillin and pesticides, either.

Please note that we have summarized the studies that exist. None of the studies examined the possibility of interaction between fumagillin and chemicals which are not typically applied in hive. As indicated in the tables and footnotes, many studies included multiple treatment groups which, for brevity, we have not described in detail. Some studies crossed fumagillin treatments with other factors, and a few of these considered other in-hive chemicals. We have reported results for each treatment sub-group in the tables; if a significant interaction had been detected, it would be evident there. Somewhat surprisingly, recent studies have not considered possible interactions between fumagillin and other chemical treatments; the major author on this subject was T.A. Gochnauer (references [27, 30, 31, 36, 37, 45])

358-363 "the effects of fumagillin was clearly postive".

In my opinion the authors did not test for the effects of fumagillin on colonies. They reviewed papers that stated that fumagillin treated colonies had certain phenotypes compared to untreated colonies. This does NOT mean that fumagillin per se has a positive effect on colonies.

See the general comments above. Please note that only controlled trials were accepted for this review, and we feel it is entirely appropriate to indicate a treatment effect is “positive” when the clear majority of field trials support such a result. Such a statement could be called into question if studies produced conflicting results, however, while we found some variability, we did not find this to occur.

Questions that I had were:

What were the differences in effects of fumagillin when applied in spring versus applications in winter, what is the current recommendation based on this paper?

We could not reach an evidence-based conclusion about the optimal timing of application, largely because there are no studies directly comparing the long term effects of fall-only applications to those of spring-only applications on overwintered colonies. Study [67] (Punko et al) almost meets this description (except they used nucleus colonies); but the doses were exceptionally low and the spring treatment was applied very late. The subject of application time is discussed at lines 598-605, as follows:

The optimum timing of fumagillin application is unclear. Most trials of fumagillin have been conducted in temperate regions with distinct spring and fall periods and clear seasonal patterns of nosema infection. Positive effects have been detected more frequently in studies that included a spring treatment, but this is not conclusive. Before 1980, spring treatments were mainly used with package bees or other special situations, such as queen mating nucs. Since 1980, most studies have included a fall or winter treatment, and some have also included a spring treatment. Studies with multiple treatment periods have usually reported colony level benefits (Tables 4-6).”

What are the minimally effective doses that are recommended to be used based on this paper?

We do not provide a recommendation to modify the label dose, and we do not think the available evidence supports a modification, not because one is not necessary but because the appropriate multi-dose studies have not been carried out. The data demonstrate that fumagillin is beneficial as currently applied, but not that the dose, timing, or application process is optimal. There are no data to adequately support any particular recommendation for colonies in spring other than the 100 mg dose with package bees. The fall label dose may have been set lower than optimal from the outset because of concern about the cost of treatment. There is no label dose specific to overwintered full-sized colonies in spring, even though application at that time is common. Early authors seem to have assumed that overwintered full sized colonies would be better treated in fall than in spring, and this does not appear ever to have been seriously challenged.

Findings related to dose are discussed at length in section 4.4 of the manuscript. We have also added to the conclusions a very brief description of an experimental strategy that would be appropriate for the development of dose and threshold recommendations.

Reviewer 2 Report

Comments and Suggestions for Authors

Dear Authors,

Your analysis of the literature data on the use of fumagillin was performed at a high level and is of great importance for beekeeping. I have a few comments.

1. A brief description of fumagillin from the point of view of chemistry and mechanism of action would enhance the Introduction, since it is associated with the feasibility and limitations of using this drug in beekeeping practice.

2. In the title and text, the name of the bee disease should be written either “Nosema disease” (with a capital letter) or “nosematosis”.

3. Line 66. А question mark is required at the end of the sentence.

4. Lines 60-66. Double numbering of questions considered in the review is used.

5. Lines 72-89. Double numbering of stages of literature search is used.

6. In Сonclusion, it would be appropriate to indicate the strict conditions and stages of further field testing of fumagillin, which would allow optimizing its use in beekeeping practice.

In addition, there are a number of errors in the References:

1. Line 680. It is need to put in a full stop after “(Hymenoptera, Apidae)”.

2. Line 691. The initials of the author Selvisabhanayakam are missing. It is correct “Selvisabhanayakam, S.”

3. Line 763. There is no comma after Meana.

4. Line 798. There is no comma after surname and a full stop after the initials of the authors “Punko RN; Currie RW; Nasr ME; Hoover SE”.

5. Line 823. It is need to put in a full stop after “Canada”.

6. Line 832. There is no comma after surname of the authors “Smart M.D.; Sheppard W.S.”

Author Response

Comments: Reviewer Two

Your analysis of the literature data on the use of fumagillin was performed at a high level and is of great importance for beekeeping. I have a few comments.

Thank you.

1. A brief description of fumagillin from the point of view of chemistry and mechanism of action would enhance the Introduction, since it is associated with the feasibility and limitations of using this drug in beekeeping practice.

We have added the following at lines 41-47:

Fumagillin is produced by the fungus Aspergillus fumigatus and targets the methionine aminopeptidase type 2 (MetAP-2) protein. Fumagillin acts against microsporidian and mammalian MetAP-2 enzymes and this is the cause of its potential toxicity to humans [3]. Fumagillin treatment stops active nosema infections but does not prevent re-infection by spores remaining in the colony, and consequently treatments are typically applied once or twice per year, outside of the honey production season [3].”

2. In the title and text, the name of the bee disease should be written either “Nosema disease” (with a capital letter) or “nosematosis”.

The phrase "nosema disease" has been changed to "Nosema disease" throughout.

3. Line 66. А question mark is required at the end of the sentence. Line 72 as revised. Corrected.

4. Lines 60-66. Double numbering of questions considered in the review is used.

5. Lines 72-89. Double numbering of stages of literature search is used.

The double numbering problem arose from formatting changes between submission and review and has been corrected.

6. In Сonclusion, it would be appropriate to indicate the strict conditions and stages of further field testing of fumagillin, which would allow optimizing its use in beekeeping practice.

We are uncertain what the reviewer intends here. If he/she is referring to regulations about the conduct of experiments, the answer will depend on the jurisdiction. In Canada, fumagillin DCH is legal to use in beekeeping and does not require a veterinarian's prescription. Tests of a higher-than-label dose, or of application during the honey production season, would present concerns; the hive products might have to be discarded or residues measured before consumption. In Europe, where routine use of this product is not permitted, conditions of testing will be more restrictive.

Instead, we have interpreted the reviewer's comment as a request that we expand upon the brief statement already included in the Conclusions (at line 653 of the revised document), which reads as follows:

However, details of dose, timing, and treatment threshold have not been optimized and more work would be appropriate in these areas.”

We have added the following at lines 655-661:

An appropriate strategy would be to establish a clear relationship between a particular measure of nosema - for example, prevalence among older bees at the end of winter - and a particular colony level outcome - for example, honey production during the following season. Subsequently, treatment strategies could be compared based on their ability to disrupt this relationship.”

In addition, there are a number of errors in the References: We thank this reviewer for his/her careful proofreading.

1. Line 680. It is need to put in a full stop after “(Hymenoptera, Apidae)”. Corrected. Line 716.

2. Line 691. The initials of the author Selvisabhanayakam are missing. It is correct “Selvisabhanayakam, S.” The citation as given (with no initial) matches the author list on my copy of the document; however, I see that Google Scholar agrees with the reviewer. Corrected. Line 727.

3. Line 763. There is no comma after Meana. Corrected. Line 799.

4. Line 798. There is no comma after surname and a full stop after the initials of the authors “Punko RN; Currie RW; Nasr ME; Hoover SE”. Corrected. Line 834.

5. Line 823. It is need to put in a full stop after “Canada”. Corrected. Line 859.

6. Line 832. There is no comma after surname of the authors “Smart M.D.; Sheppard W.S.” Corrected. Line 868.

Reviewer 3 Report

Comments and Suggestions for Authors

I appreciate the opportunity to evaluate this systematic review. This article presents a review on the literature evidences of fumagillin field trials for the treatment of nosema disease in honey bee colonies. It brings contribution in the field, although I do not see an exceptional and additional impact.

The impact of this antibiotic's use in beekeeping on human health and the environment is not given enough consideration. Thus, I propose including at least a sentence or two in the study that offer a more contemporary, serious scientific perspective. Otherwise, it will appear that the study is endorsing the antibiotic's use. My opinion is based on the fact that fumagillin is registered for use in Canada to treat nosemosis, while it is sold as the salt dicyclohexylamine (DCH) in commercial products. This chemical is genotoxic and carcinogenic, and investigations on rats have shown that it is five times more toxic than fumagillin. As a result, there may be health risks to humans from it.

You indicate that studies using bees kept in cages were not included. To my knowledge and based on the literature, fumagilin appeared to have the highest amount of toxicity in this sort of experiment. I respect your study design in which you did not include these results into account, however, since this is a „systematic review“, maybe you should think about including at least a sentence or two regarding cage experiments.

The conclusions and recommendations for further studies are properly addressed. The listed references are appropriate, however the style of citing in the manuscript is incorrect, which I addressed in comments bellow.

In my opinion, minor corrections (listed in the text above and comments bellow) are needed before it can be published:

Line 41: Misspelling error: N, ceranae

Lines 60-66: Sequence numbers were written two times, please correct this.

Lines 70-89: Please delete sequence numbers (4, 5, 6, 7).

Line 216: The reference Girardeau is missing the reference number next to the name of the author, please correct this. This comment applies for the same throughout the manuscript (e.g. Lines 235, 239 etc.).

Author Response

Comments: Reviewer Three

I appreciate the opportunity to evaluate this systematic review. This article presents a review on the literature evidences of fumagillin field trials for the treatment of nosema disease in honey bee colonies. It brings contribution in the field, although I do not see an exceptional and additional impact.

We thank the reviewer for his or her time and attention.

The impact of this antibiotic's use in beekeeping on human health and the environment is not given enough consideration. Thus, I propose including at least a sentence or two in the study that offer a more contemporary, serious scientific perspective. Otherwise, it will appear that the study is endorsing the antibiotic's use. My opinion is based on the fact that fumagillin is registered for use in Canada to treat nosemosis, while it is sold as the salt dicyclohexylamine (DCH) in commercial products. This chemical is genotoxic and carcinogenic, and investigations on rats have shown that it is five times more toxic than fumagillin. As a result, there may be health risks to humans from it.

We have added the following under the section entitled 4.5. Toxicity concerns:

The dicyclohexylamine counter-ion appears to be both more persistent in hive products and more toxic to mammals than fumagillin itself, and may be a cause of concern [3].”

We agree with the reviewer that, in principle, exposure to fumagillin in food could constitute a human health risk and that regulatory authorities should carefully assess that risk. We further agree that the toxicity and stability of dicyclohexylamine (both for bees and mammals) is probably greater than that of fumagillin, and are tentatively inclined to support a suggestion previously made by van den Heever et al. that dicyclohexylamine should be replaced with a less problematic counter-ion. We would also agree that widespread antibiotic use can have negative effects on the environment, including the suppression of beneficial microorganisms and the development of resistant strains of pathogens. Indeed, the possibility of resistance to fumagillin was one of our research questions (Question 2 of the list in the introduction), and the discussion section of the paper begins by considering that point.

We had seriously considered including a more extensive discussion of the literature related to fumagillin toxicity, but felt in order to do so would add considerable length to this review and move it away from its intended focus. Moreover, we point out that these concerns have already been raised in another relatively recent review by van den Heever et al. (2014) [3]. We feel section 4.5 does outline concerns with toxicity, and we do not mean to suggest that using fumagillin to treat bees is without potential concern. Moreover, maximum residue limits in countries like Canada do exist to protect the consumer from high levels of fumagillin in honey.

You indicate that studies using bees kept in cages were not included. To my knowledge and based on the literature, fumagilin appeared to have the highest amount of toxicity in this sort of experiment. I respect your study design in which you did not include these results into account, however, since this is a „systematic review“, maybe you should think about including at least a sentence or two regarding cage experiments.

We have added the following at line 494:

Cage-based experiments play an important role in honey bee science because they can allow more sensitive measurements of phenomena which are difficult to detect in the field. However, the extrapolation of lab studies to the field is rarely straightforward.”

Many cage studies were examined in the course of this work, primarily to determine whether they included a field component, and also in the hope of identifying something analagous to a minimum inhibitory concentration for either species of Nosema. Some papers did include both a field component and a lab component, although we only summarized the field component. Tom Webster's brief product comparison, [70], comes to mind, although there were certainly others. We have discussed Huang et al's [73] very prominent cage study at some length (lines 480-508) because it makes claims about the response of Nosema ceranae to fumagillin. In addition, we cite cage studies by Standifer and Furgala [74] and several studies by van den Heever et al. in relation to the questions about fumagillin toxicity to bees (line 516; and section 4.5).

The conclusions and recommendations for further studies are properly addressed. The listed references are appropriate, however the style of citing in the manuscript is incorrect, which I addressed in comments bellow.

In my opinion, minor corrections (listed in the text above and comments bellow) are needed before it can be published:

Line 41: Misspelling error: N, ceranae Now at line 49. Corrected.

Lines 60-66: Sequence numbers were written two times, please correct this. This problem was not present on the draft we sent, and seems to have been introduced through automated formatting changes. It has been corrected, although since we are uncertain of the cause, we are also uncertain whether it will recur.

Lines 70-89: Please delete sequence numbers (4, 5, 6, 7).

See comment above.

Line 216: The reference Girardeau is missing the reference number next to the name of the author, please correct this. This comment applies for the same throughout the manuscript (e.g. Lines 235, 239 etc.).

We take it that the reviewer wishes us to give the article reference immediately after the author's name, rather than at the end of the sentence. We have made this change for the cases we identified, namely:

Girardeau (line 223)

Szabo & Heikel (line 242)

Wyborn & McCutcheon (line 246)

Goodman et al. (line 248)

Woyke (line 387)

Goodman et al (line 389)

Wyborn & McCutcheon (line 623)

Reviewer 4 Report

Comments and Suggestions for Authors

It is rare for me to review a manuscript with no flaws. The manuscript is well composed, thorough, important and timely.  It could certainly be published without any changes. However, the authors might consider making more recommendations for future research on the effects of fumagillin on honey bees.  For example, not much has been published on the effects of fumagillin on queen or drone bees.  Perhaps there are subtle effects on queen and drone fertility and longevity.

Author Response

It is rare for me to review a manuscript with no flaws. The manuscript is well composed, thorough, important and timely. It could certainly be published without any changes.

We are delighted to receive such a positive response!

However, the authors might consider making more recommendations for future research on the effects of fumagillin on honey bees. For example, not much has been published on the effects of fumagillin on queen or drone bees. Perhaps there are subtle effects on queen and drone fertility and longevity.

We notice that reviewer two expressed a similar sentiment, and have added the following to the conclusions section:

An appropriate strategy would be to establish a clear relationship between a particular measure of nosema - for example, prevalence among older bees at the end of winter - and a particular colony level outcome - for example, honey production during the following season. Subsequently, treatment strategies could be compared based on their ability to disrupt this relationship. In addition, there is very little information about the effects of treatment on queen survival, and studies that report effects on colony survival have usually been under-powered.”

Effects on queens and drones are certainly possible. Shimanuki (in work not cited in the manuscript) did some work on queens many years ago, but it was mainly to establish whether there was a risk of nosema transmission in queen cages, and was not a trial of fumagillin. Moeller [50] summarized data related to the risk of nosema infection in queens; he believed that fumagillin reduced supersedures, but the question does not seem to have been directly tested. Fertility was not examined. Given that the commercial product slightly reduces the lifespan of uninfected workers (in cages; there are no comparable data from colonies), an effect on queen and drone longevity would be expected; however this effect is presumably outweighed by the benefit of not acquiring nosema. Many have noticed that queen longevity isn't what it used to be and the arrival of a second nosema species remains a plausible explanation. Early authors such as Farrar and Moeller were convinced that nosema was a major cause of queen death.

Round 2

Reviewer 1 Report

Comments and Suggestions for Authors

Thank you for the extensive answers to my comments. I appreciate the effort they put in to explain their reasoning. I can see the issues the authors found in comparing studies with different designs in  a meta-analysis. 

I really like the supplementary figures, in my eyes, they contribute a lot to the message the authors convey.

I have a few minor edits, to weaken some of the very strong statements in this manuscript, only.

Line 14

Fumagillin treatment is safe for honey bee colonies

Replace with

We found no negative effects of fumagillin treatments on honey bee colonies.

From this review, this cannot be concluded. This sentence needs to be erased from the manuscript, in my view.

Line 15

There were no reports of significant harmful effects.

Change to

We reviewed fifty publications about fumagillin that met our criteria as controlled field trials and found no reports of significant harmful effects on honey bee colonies.

Line 24

Furthermore, fumagillin appears to be safe for honey bee colonies.

According to the fifty studies we reviewed, fumagillin appears to be safe for honey bee colonies.

Or

We found no negative effects of fumagillin in our study.

Line 25

No study showed harm to any measure of colony health.

Replace with

Our study showed no negative effects on colony health.

644-646

fumagillin has been widely used by ...with no cases of harm to humans or bees cited in the beekeeping literature.

change the second part to

and we found no cases of harm to humans or bees cited in the fifty publications we reviewed.

Reason: Cases of harm of fumagillin to humans might be reported in medical journals, not beekeeping ones.

652-653l

No controlled study has detected harm to the honey bee colony from fumagillin use, and ...

Change to

None of the fifty studies we reviewed reported harm to the ...

Author Response

In response to your comments, we have made the following changes to the manuscript:

Changes to Manuscript 10 Dec 2023

Lines 14 – 16 (Line numbers for corrected ms).

Original:

“Furthermore, fumagillin treatment is safe for honey bee colonies. There were no reports of sig-nificant harmful effects and many reports of improvements in colony survival, size, and produc-tivity.”

Change:

“We also found no significant negative effects of fumagillin treatments on honey bee colonies, for the field trials meeting our criteria for review. There were also many reports of improvements in colony survival, size, and productivity with treatment.”

Line 25-26:

Original:

“Furthermore, fumagillin appears to be safe for honey bee colonies. No studies showed harm to any measure of colony health.”

Change:

“Furthermore, our study showed no negative effects on colony health.”

645-647

Original:

“fumagillin has been widely used by ...with no cases of harm to humans or bees cited in the beekeeping literature.”

Change:

“fumagillin has been widely used by ...and we found no cases of harm to humans or bees cited in the fifty field publications reviewed.”

653-654

Original:

“No controlled study has detected harm to the honey bee colony from fumagillin use, and ...

Change:

“None of the studies we reviewed reported harm to the ...”